# Robotic Bronchoscopy in Lung Cancer Diagnosis

**DOI:** 10.3390/cancers16061179

**Published:** 2024-03-17

**Authors:** Vasileios S. Skouras, Ioannis Gkiozos, Andriani G. Charpidou, Konstantinos N. Syrigos

**Affiliations:** 1Department of Pulmonary Medicine, 401 General Military Hospital, 1 Kanellopoulou Avenue, 11525 Athens, Greece; 2Oncology Unit, 3rd Department of Internal Medicine, National and Kapodistrian University of Athens, 152 Μesogeion Avenue, 11527 Athens, Greece; yiannisgk@hotmail.com (I.G.); dcharpidou@yahoo.gr (A.G.C.); ksyrigos@med.uoa.gr (K.N.S.)

**Keywords:** robotic bronchoscopy, guided bronchoscopy, pulmonary nodule

## Abstract

**Simple Summary:**

The widespread use of chest CT has increased the number of detected pulmonary nodules. Indeterminate nodules of intermediate risk for lung cancer warrant further evaluation with PET-CT or sampling with transthoracic needle biopsy or bronchoscopy. The diagnostic yield of conventional bronchoscopy is limited by the inability to reach distal airways. Robotic bronchoscopy (RB) is a novel bronchoscopic technique that aims to overcome this limitation. The aim of this review was to provide the rationale behind the need for RB use in clinical practice, describe RB procedure, and summarize data regarding its efficacy and safety.

**Abstract:**

Background: The widespread use of chest CT has increased the number of detected pulmonary nodules. Nodules with intermediate risk of malignancy warrant further evaluation with PET-CT or sampling. Although sampling with conventional bronchoscopy presents lower complication rates compared to transthoracic needle biopsy (TTNB), it is limited by the inability to reach distal airways. To overcome this shortcoming, a new bronchoscopic technique named robotic bronchoscopy (RB) has emerged. Methods: A literature review was used to clarify the rationale behind RB emergence, describe RB procedure, and summarize data regarding its efficacy and safety. Results: The FDA has approved three RB platforms for clinical use. RB is safe, presenting a mortality and complication rate of 0% and 0–8.1%, respectively. Common complications include pneumothorax (0–5.7%) and minor bleeding (0–3.2%). However, its diagnostic yield remains lower than that of TTNB. Conclusions: RB is a promising bronchoscopic technique that aims to overcome the limitations of conventional bronchoscopy and improve upon the current techniques of guided bronchoscopy for the investigation of pulmonary nodules. Despite the lower complication rate, current evidence suggests a lower diagnostic yield compared to TTNB. Additional studies are required to adequately evaluate the role of RB in the diagnosis of pulmonary nodules.

## 1. Introduction

Lung cancer ranks among the most common malignancies and represents the most lethal cancer, responsible for 20% of cancer-related deaths worldwide [1,2]. Its dismal prognosis is closely related to the disease being frequently (80%) disseminated at the time of diagnosis, as reflected by the significant difference (up to 90% versus up to 40%) in the 5-year survival rates between early- and advanced-stage disease [1,3,4]. This is why efforts to improve lung cancer survival have focused on the timely detection of early disease, which provides the opportunity for complete tumor resection and offers higher probability of cure [5]. Screening high-risk populations (i.e., smokers or ex-smokers aged more than 50–55 years old) with low-dose chest CT has been found to reduce lung cancer mortality by at least 20% [6,7]. As a consequence, low-dose CT screening programs have been implemented in several countries.

The implementation of lung cancer screening, along with the widespread use of chest CT for diagnostic purposes, has significantly increased the number of detected pulmonary nodules [8]. In the US, around 5 million chest CT scans are performed and 1.5 million pulmonary nodules are detected each year [9]. The etiology of pulmonary nodules is extensive and includes a variety of non-malignant (i.e., infections, inflammatory diseases, vascular disorders, congenital malformations, benign tumors, etc.) entities along with malignant tumors, lung cancer being the most common among them [10]. The extensive differential diagnosis, along with the low (up to 5%) prevalence of malignancy among detected pulmonary nodules, makes the management of such lesions rather challenging. The greatest challenge is to discriminate the nodules warranting further investigation and, potentially, surgical excision to cure an underlying malignancy (minority) from those that are non-malignant and may require no intervention (majority) [9,10,11,12,13].

The identification of the nodules requiring further investigation is mainly based on nodule radiographic characteristics (e.g., size, location, margins, density, growth rate, ^18^FDG uptake) and patient demographic data (e.g., age, sex, smoking exposure, presence of emphysema, history of cancer) [10,11,12,13]. Pulmonary nodules lacking radiographic characteristics with high specificity for non-malignancy (i.e., stable or reduced size during a 2-year period, benign pattern of calcification, intranodular fat, etc.) are termed indeterminate [11,12,13]. The management of indeterminate pulmonary nodules requires initial assessment of their clinical probability of lung cancer. For this purpose, guidelines suggest the use of prediction models (i.e., Mayo Clinic model and Brock/Herder model) that categorize nodules into low- (probability <5% or <10%, respectively), intermediate- (probability 5–65% or 10–70%, respectively) or high-risk (probability >65% or >70%, respectively) ones [13,14]. Although low- and high-risk nodules are usually managed with watchful waiting or surgical excision/biopsy, respectively, intermediate-risk nodules are typically considered for further evaluation with PET-CT or sampling [10,13]. During the last few years, a new bronchoscopic technique for pulmonary nodule sampling has emerged, namely robotic bronchoscopy.

The aims of this review were to provide the rationale behind the need for another bronchoscopic technique, to describe the new technique, and to summarize data regarding its efficacy and safety. To accomplish these aims, a search in the PubMed database was performed in October 2023 with the use of the following keywords: “robotic bronchoscopy”, “robotic-assisted bronchoscopy”, and “guided bronchoscopy”. All English-language articles describing the platforms and the technique of robotic bronchoscopy, as well as those providing data regarding its diagnostic yield and complication rate, were used for this purpose.

## 2. Rationale for the Use of Robotic Bronchoscopy

The available methods to approach and sample pulmonary nodules are generally classified into surgical and non-surgical or minimally invasive. The surgical methods comprise video-assisted thoracoscopic surgery (VATS) and open thoracotomy, while the non-surgical ones comprise CT-guided transthoracic needle biopsy (TTNB), conventional bronchoscopy with endobronchial and/or transbronchial biopsy, guided bronchoscopy with transbronchial biopsy, and bronchoscopic transparenchymal nodule access (BTPNA) [10,13,15,16]. Surgical methods are usually used to sample pulmonary nodules with high clinical probability of malignancy in patients with average surgical risk because, in such cases, intraoperative confirmation of malignancy can be followed by curative lobectomy [10,13]. In contrast, the presence of high surgical risk or intermediate clinical probability of malignancy usually prompts the use of non-surgical sampling methods. Despite the overlap in the clinical usefulness of these methods, the selection of the most appropriate non-surgical sampling method in such cases is generally based on nodule location. According to their location, pulmonary nodules are divided into central (i.e., those lying in the inner third of the lung) or peripheral (i.e., those lying in the middle and outer third of the lung) [10]. Peripheral nodules can be more easily sampled with TTNB or guided bronchoscopy, while central nodules can be adequately (i.e., sensitivity of 88% for malignancy) approached with conventional or guided bronchoscopy [10,15]. Requirement for mediastinal staging, such as in the case of centrally located lesions, may also influence the decision for the most appropriate non-surgical sampling method in favor of bronchoscopy, since TTNB does not provide this opportunity [17]. Considering the low (12%) prevalence of centrally located nodules, however, the great challenge in the diagnostic approach of pulmonary nodules is the sampling of the peripheral ones.

Selection of the most appropriate method to sample peripheral pulmonary nodules should generally take into account nodule location (e.g., nodules lying in the outer or middle third of the lung may be more easily approached with TTNB or guided bronchoscopy, respectively), the method’s characteristics (i.e., efficacy and safety), and the method’s availability. Although TTNB presents high diagnostic accuracy (sensitivity: 90%, specificity: 97%) for this purpose, it is associated with a 15% and 1% risk for pneumothorax and bleeding, respectively [10,15,18]. In contrast, transbronchial biopsies through conventional bronchoscopy present significantly lower complication rates (1.5% pneumothorax, 0.5% bleeding) [10]. The diameter of conventional bronchoscopes, however, prevents their advancement beyond the subsegmental bronchi, precluding thereby sampling of peripheral lesions under direct visualization. Accordingly, the sensitivity of conventional bronchoscopy in the diagnosis of malignant peripheral pulmonary nodules does not exceed 50% and decreases further when a specific benign diagnosis is sought [10]. To overcome this limitation, ultrathin bronchoscopes and bronchoscopic guidance technologies, including radial-endobronchial ultrasound (r-EBUS), virtual bronchoscopy (VB), and electromagnetic navigation (EMN), were developed [10,11]. Of special interest is navigational bronchoscopy (i.e., VB and EMN), which uses CT images to create a virtual 3-D map of the airways, providing pathway options to the target lesion [10]. Despite the sophisticated technologies, the diagnostic yield of guided bronchoscopy was suboptimal (70%) in a 2012 meta-analysis [19]. This has been mainly attributed to the continuing inability of newer bronchoscopes to reach distal airways and to the occasional occurrence of false navigation due to the phenomenon of “CT-to-body divergence” (i.e., discrepancy between electronic virtual target and actual anatomic location of the target lesion), which may be caused by respiratory movements, focal atelectases, or anatomic changes during the time from CT to bronchoscopy [20,21]. To overcome these limitations, a new bronchoscopic technique has been introduced for the investigation of pulmonary nodules, which combines ultrathin robotically propelled bronchoscopes with endobronchial guidance and is called robotic bronchoscopy (RB).

The emergence of this technique can be considered the result of efforts to improve the diagnostic yield of guided bronchoscopy in the evaluation of pulmonary lesions that cannot be approached with conventional bronchoscopy. Therefore, patients with peripheral pulmonary nodules of intermediate risk of malignancy would be ideal candidates for robotic bronchoscopy. However, the high diagnostic accuracy of TTNB for the same purpose requires robotic bronchoscopy to be more accurate than and at least as safe as TTNB before being implemented in routine clinical practice.

## 3. Robotic Bronchoscopy Procedure

A schematic room setup for the robotic bronchoscopy procedure with various platforms is shown in Figure 1. RB is typically performed with the patient under general anesthesia and intubated with an indwelling endotracheal tube [20,22]. However, a patient who underwent the procedure under moderate sedation has also been described [23]. Prior to the insertion of the robotic bronchoscope, conventional white-light bronchoscopy is performed to inspect the airways for central lesions and clear the airways from secretions [20]. Then, the robotic bronchoscope is docked to the endotracheal tube and the bronchoscopist drives the bronchoscope to the main carina and along the main bronchi to accomplish registration (i.e., correlation of the live bronchoscopic view with the CT-based virtual reconstruction of the airways) [24]. Following registration, the bronchoscopist follows the preplanned pathway to reach the target lesion. Once the target is reached, confirmation of the bronchoscope’s position is usually required with the use of ancillary imaging modalities, such as r-EBUS, fluoroscopy, or cone beam CT [20,24,25]. After confirming successful navigation to the target lesion, the bronchoscope is fixed to a steady position and sampling tools (e.g., needle, forceps, cryobiopsy probe) are passed through the working channel for biopsies to be obtained [20,24,26]. Although each robotic platform has its own needle (e.g., Auris needle and Flexision needle) for nodule sampling, other needles (e.g., 21G Olympus TBNA needle, 19G or 21G Intuitive Surgical needle) and biopsy tools (e.g., Endojaw Olympus forceps, 1.1 mm Erbe Tuebingen cryoprobe) can also be used [20,24,26]. Following adequate tissue acquisition, the robotic bronchoscope is retracted and clearance of the airways from secretions with the conventional bronchoscope may be required again. During the procedure, the patient is mechanically ventilated with a tidal volume of approximately 8 mL/kg ideal body weight and a positive end-expiratory pressure (PEEP) of 8–10 cm H_2_O [20,22,27]. Neuromuscular blockade may rarely be used to suppress patient movements and cough and maintain a steady field during navigation and sampling [20,27]. The mean navigation and procedure time ranges from 5 to 21 min and from 36 to 78 min, respectively, and has been shown to decrease with increasing experience [22,24,28]. In a study aiming to determine the learning curve of the procedure, 18 cases were required for a certain bronchoscopist to achieve a stable procedure time [29].

## 4. Robotic Bronchoscopy Platforms

Until recently, two robotic platforms were approved by the FDA, namely, the Monarch (in 2018) and the Ion Endoluminal (in 2019) System by Auris Health Inc. (Redwood City, CA, USA) and Intuitive Surgical (Sunnyvale, CA, USA), respectively [24]. In 2023, FDA clearance was also granted to the Galaxy System by Noah Medical (San Carlos, CA, USA) [30,31]. Table 1 shows a comparison of the technical characteristics of these platforms.

### 4.1. Monarch System

Monarch consists of a bronchoscope system, cart, and tower (Figure 2) [20]. The bronchoscope system consists of an outer sheath (6.0 mm in diameter) and an inner bronchoscope (4.2 mm outer diameter) with four-way steering control. The bronchoscope contains a camera that provides continuous visualization during navigation and sampling, an integrated light source, a working channel of 2.1 mm in diameter, and a separate suction channel. The cart accommodates the electronic systems for the power of the platform and two robot arms that actuate the drive cables of the bronchoscope. The tower has two computers that run the platform and an integrated monitor that displays real-time video from the bronchoscope camera in conjunction with information from the robotic system. A two-joystick handheld controller that allows the bronchoscopist to drive the bronchoscope is also part of the tower. The system uses EMN guidance with an external electromagnetic field generator and reference sensors to navigate the bronchoscope to the target lesion [20,32]. A pre-procedural thin-cut chest CT is required for a 3-D image reconstruction to be generated and pathways to target lesion to be mapped and uploaded to the robot. During the procedure, the bronchoscopist navigates the bronchoscope with the use of the controller to the target lesion according to automatically or manually generated pathways. Throughout navigation, continuous feedback is provided regarding the location and the distance from the target. Once the target is reached, successful navigation can be confirmed with r-EBUS or fluoroscopy and sampling with the advancement of tools through the working channel can be performed [20,24,32].

### 4.2. Ion Endoluminal System

The Ion platform (Figure 2) consists of a robotic system cart, a controller, and a single ultrathin bronchoscopic catheter that has 3.5 mm outer diameter with 2.0 mm working channel and can articulate 180 degrees. The catheter is controlled with a trackball and scroll-wheel that advances and retracts the catheter and drives it into the bronchial tree [28,33]. The movement of the catheter is enabled by the robotic system cart through an arm that responds to input from the controller and a pull-wire system that allows catheter driving into the airways under direct visualization with the use of a vision probe. Two monitors placed at the top of the cart display virtual and live airway views while providing information about the target lesion and the position of the catheter in the bronchial tree. The Ion system uses shape-sensing technology for guidance. This is accomplished via a thin flexible fiber embedded throughout the entire length of the bronchoscopic catheter, which measures its own shape hundreds of times per second, providing continuous depiction of the shape and position of the catheter in the airways during the procedure [28]. A pre-procedural thin-cut (0.75–1.25 slice thickness) chest CT is required for a virtual airway map to be generated with the use of the integrated PlanPoint software (https://www.intuitive.com/en-us/products-and-services/ion/planpoint-software accessed on 6 February 2024) and pathways to the target lesion to be created [24]. The virtual map is registered with patient’s airway anatomy with the use of a vision probe that is inserted through the bronchoscopic catheter into the bronchial tree to provide live endoscopic view that correlates with the virtual images. Navigation to the target is performed with the vision probe in place. During the procedure, the shape-sensing technology transmits information about shape and motion changes of the catheter to the computer, which correlates it with information from the chest CT to determine the catheter’s position and its distance from the target [24]. Once the target is reached, successful navigation is confirmed by r-EBUS and/or fluoroscopy. Then, the vision probe is removed to allow the insertion of a flexible needle (Flexision) for tissue acquisition [24].

### 4.3. Galaxy System

The Galaxy System (Figure 2) consists of a cart, a single-use bronchoscopic catheter (4.0 mm in outer diameter with a 2.1 mm working channel) with an integrated camera that provides continuous real-time visualization of the airways, and a two-joystick controller. The cart accommodates the system’s computer and bears a monitor and an instrument arm responsible for the catheter’s actuation. Following 3-D reconstruction of the airways from a pre-procedural chest CT, pathway mapping and guidance to the target lesion is provided through EMN [21,30]. Once the target lesion is reached, correct position of the catheter is confirmed with the tool-in-lesion technology (TiLT), which combines EMN with digital tomosynthesis and augmented fluoroscopy to diminish “CT-to-body divergence”. Biopsies can then be obtained with sampling tools advanced through the working channel of the catheter [30].

## 5. Feasibility and Efficacy of Robotic Bronchoscopy Platforms

Great variability exists regarding the appropriate metric for expressing the efficacy of RB. Although the sensitivity and specificity of the technique for the detection of malignancy is used by some researchers, diagnostic yield represents the most widely used metric for this purpose [8]. Even in this case, though, the inconsistency regarding the calculation of diagnostic yield across various studies renders the comparison of their results rather challenging. Indeed, Vachani et al. showed that the use of strict (i.e., malignant plus specific benign diagnoses), intermediate (i.e., strict plus non-specific benign findings), or liberal (i.e., intermediate plus nondiagnostic cases) criteria to define diagnosed cases may produce significant variations in the calculated diagnostic yields, which could possibly affect the interpretation of the study’s results [34]. In this review, diagnostic yields with the use of strict criteria are reported (unless otherwise indicated).

The first study (REACH) demonstrating the navigational capabilities of Monarch was published in 2018 by Chen et al., who used human cadavers to compare the robotic platforms with conventional bronchoscopes in terms of how far in the distal airways they could reach [35]. Despite the same diameter (i.e., 4.2 mm), the robotic bronchoscope presented better maneuverability and was able to navigate further in the distal airways (9th generation versus 6th generation bronchi). This superiority of the Monarch bronchoscope was attributed to its special design (i.e., mother–daughter configuration), which allows the outer sheath to provide support against the bronchial walls in order for the inner bronchoscope to navigate further into the bronchial tree [20]. A year later, the ACCESS study evaluated the diagnostic yield of Monarch in human cadavers with implanted peripheral (1.6 cm mean distance from the pleura) artificial tumor targets of 1–3 cm in diameter [36]. The high (97%) diagnostic yield of RB in this study, however, was considered rather optimistic due to the absence of CT-to-body divergence from the use of cadavers. A retrospective multicenter study of 167 lesions (mean diameter of 25 mm), 71% of which were located in the outer third of the lung, reported navigation success rate (i.e., tissue diagnosis or r-EBUS confirmation) and overall diagnostic yield of 88.6% and 69%, respectively [37]. During the last 4 years, several retrospective studies assessing the effectiveness of Monarch have reported a diagnostic yield of 36–62% (Table 2) [37,38,39,40,41]. More reliable data, however, come from the multicenter prospective study, BENEFIT, which assessed the diagnostic yield of Monarch in 54 human patients with peripheral pulmonary lesions of 1–5 cm [42]. Despite successful lesion localization in 96.2% of the cases (as confirmed by r-EBUS), the overall diagnostic yield of the technique did not exceed 67%. In a retrospective study of 124 nodules investigated with Monarch, nodule size > 2 cm and r-EBUS confirmation was associated with higher diagnostic yield, the latter remaining unaffected by nodule location or bronchus sign [40].

The first feasibility study for Ion was published in 2019 by Fielding et al., who evaluated 29 pulmonary lesions and reported navigation success rate and diagnostic yield of 96.6% and 79.3%, respectively [43]. In 2020, Yarmus et al. compared Ion RB to EMN and ultrathin bronchoscope with r-EBUS in a human cadaver study and found that RB presented a higher (80% versus 45% versus 25%) tool-in-lesion rate (as confirmed by cone beam CT) [44]. Several retrospective studies during the last 4 years confirm the high (77.6–100%) navigation success rates of Ion and report a diagnostic yield of 63–81% (Table 2) [25,26,45,46,47,48]. Moreover, a single-center prospective study evaluating Ion in the investigation of 59 pulmonary nodules reported a navigation success rate and diagnostic yield of 85% and 64%, respectively [27]. The first multicenter prospective study (i.e., PRECIsE) evaluating Ion in the diagnostic approach of peripheral pulmonary nodules is still ongoing; initial results from this study indicate a navigation success rate of 97% [28].

An animal study (MATCH) evaluating the tool-in-lesion (TIL) accuracy of Galaxy in 20 simulated pulmonary nodules implanted in four pigs reported TIL and biopsy success rates of 95% and 100%, respectively [21]. Although the first clinical trial (FRONTIER) evaluating the feasibility and safety of Galaxy in human patients is still ongoing, preliminary data from the investigation of 19 peripheral pulmonary nodules (mean diameter of 20 mm), presented at the Annual Conference of the American Association for Bronchology and Interventional Pulmonology (AABIP) in 2023, showed a navigation success rate, tool-in-lesion rate, and diagnostic yield of 100%, 100%, and 89.5–94.7%, respectively [49].

In a recent meta-analysis of 12 studies with 838 nodules, the diagnostic yield of robotic bronchoscopy with any platform was 81.9% and fell to 75.9% when only full-length articles (n = 7) with diagnostic data were included in the analysis [50]. In another meta-analysis assessing the diagnostic yield of guided bronchoscopy with the inclusion of six RB studies with 483 nodules, a similar diagnostic yield of 77.6% was also reported for RB [8].

**Table 2 cancers-16-01179-t002:** Human studies assessing the diagnostic yield and safety of robotic bronchoscopy.

Study	Country	Study Design	Platform	Lesions (n)	Lesion Size (mm)	Peripheral Location	Bronchus Sign	Position Confirmation	Successful Navigation	Navigation Time (Minutes)	Procedure Time (Minutes)	Diagnostic Yield *	Complication Rate
Rojas-Solano et al. (2018) [51]	Costa Rica	Feasibility	Monarch	15	26 (range: 10–63)	80%	100%	No	93%	21	NR	-	0%
Chaddha et al. (2019) [37]	US	Retrospective	Monarch	167	25 ± 15	71%	63.5%	Yes (r-EBUS)	88.6%	17.8 ± 19.1	58.6 ± 31.4	69.1% (77% *)	Overall: 6.0%, pneumothorax (3.6%)—chest tube (2.4%), bleeding (2.4%)
BENEFIT, Chen et al. (2021) [42]	US	Prospective (feasibility multicenter)	Monarch	54	23 (IQR: 15–29)	100%	59.3%	Yes (r-EBUS)	96.2%	13 (IQR: 13–24)	51 (IQR: 44–64)	67% (74.1% *)	Pneumothorax (3.7%)—chest tube (1.9%)
Ekeke et al. (2021) [38]	US	Retrospective	Monarch	25	Range: 8–69	64%	84%	NR	96%	NR	NR	80%	0%
Cumbo-Nacheli et al. (2022) [39]	US	Retrospective	Monarch	20	22 ± 7	90%	50%	Yes (CBCT and r-EBUS)	100%	9.8 (range: 3–41)	36.4 (range: 15–66)	65%	NR
Agrawal et al. (2023) [40]	US	Retrospective	Monarch	124	20.5 (IQR: 13–30)	45%	75%	Yes (r-EBUS in 82% of cases)	94.4%	NR	NR	65% (77% *)	Overall: 4.8%, pneumothorax (1.6%), bleeding (3.2%)
Khan et al. (2023) [41]	US	Retrospective	Monarch	264	19.3 (range: 3.2–72.5)	58.9%	30.1%	Yes (r-EBUS, fluoroscopy)	98%	NR	62.3 ± 27.2	56% (85% *)	Overall: 7.2%, pneumothorax (5.7%)—chest tube (3.8%), bleeding (1.5%)
Fielding et al. (2019) [43]	Australia	Feasibility	Ion	29	12.2 ± 4.2	-	58.6%	Yes (r-EBUS)	96.6%	NR	63.9 ± 24.4	79.3%	0%
Benn et al. (2021) [27]	US	Prospective (single-center)	Ion	59	21.9 ±11.9 (range: 7–60)	NR	46%	Yes (CBCT)	85%	NR	65 ± 25	64% (79% *)	Pneumothorax (3.8%)
Simmof et al. (PRECIsE) (2021) [28]	US	Prospective (multicenter)	Ion	67	20 (IQR: 14–27)	-	37.3%	Yes (r-EBUS, fluoroscopy)	97%	5.0 (IQR: 3–10)	66.5 (IQR: 50–85.5)	NR	Overall: 3.4%, arrhythmia (1.7%), pneumonia (1.7%)
Kalcheim-Dekel et al. (2022) [46]	US	Retrospective	Ion	159	18 (IQR: 13–27)	66.7%	62.9%	Yes (r-EBUS, fluoroscopy)	98.7%	NR	64 (IQR: 40–116)	63% (82% *)	Overall: 3.0%, pneumothorax (1.5%)
Yu Lee-Mateus et al. (2022) [48]	US	Retrospective	Ion	113	18 (13–27)	-	NR	Yes (r-EBUS)	100%	NR	78 (IQR: 62.5–92.5)	76.9% (87.6% *)	Overall: 4.4%, pneumothorax (3.5%)
Oberg et al. (2022) [26]	US	Retrospective (cryobiopsy)	Ion	120	22 (IQR: 13–3)	100%	48%	Yes (r-EBUS)	100%	NR	NR	76% (90.2% *)	Overall: 8.1%, pneumothorax (5.4%)—chest tube (2.7%), bleeding (2.7%)
Reisenauer et al. (2022) [22]	US	Prospective (single-center)	Ion	30	17.5 (range: 10–30)	-	40%	Yes (r-EBUS, fluoroscopy)	96.7%	NR	NR	76.7% (93.3% *)	Overall: 6.25% (arrhythmia, hypotension)
Styrvoky et al. (2022) [25]	US	Retrospective	Ion	209	19 (range: 7–73)	85%	60.3%	Yes (r-EBUS, CBCT)	77.6%	NR	NR	76.5% (91.4% *)	Pneumothorax (1.0%)—chest tube (0.5%)
Hammad-Altaq et al. (2023) [45]	US	Retrospective	Ion	42	12 (IQR: 10–18)	71.4%	59.5%	Yes (r-EBUS)	100%	NR	NR	81% (88.1% *)	0%
Low et al. (2023) [47]	US	Retrospective	Ion	143	17 (IQR: 12–27)	48%	40%	Yes (r-EBUS)	91.9%	NR	NR	77%	Pneumothorax (1.5%)—chest tube (1.5%)
Reisenauer et al. (2022) [52]	US	Prospective	Ion	270	18.8 ± 6.5	100%	NR	Yes (r-EBUS)	NR	NR	63 ± 30	NR	Overall: 4.1%, pneumothorax (3.3%)—chest tube (0.4%), bleeding (0.8%)
FRONTIER, Saghaie et al. (2023) ^#^ [49]	Australia	Prospective	Galaxy	19	20	NR	NR	NR	100%	NR	NR	89.5% (94.7% *)	Pneumothorax (11%)—chest tube (5%), pneumonia (5%)

*: reported diagnostic yields are calculated with the use of strict criteria, i.e., [(malignant diagnoses + specific benign diagnoses)/total procedures], while values in parentheses represent diagnostic yields with the use of intermediate criteria: [(malignant diagnoses + specific benign diagnoses + benign findings)/total procedures], #: preliminary data presented in 2023 by the AABIP (American Association for Bronchology and Interventional Pulmonology), r-EBUS: radial-endobronchial ultrasound, CBCT: cone beam computed tomography, NR: not reported.

## 6. Robotic Bronchoscopy Safety

RB seems to be safe since it presents a mortality and complication rate of 0% and 0–8.1%, respectively (Table 2). The most common complications include pneumothorax (0–5.7%), requiring chest tube placement in approximately half of the cases, and minor bleeding (0–3.2%). A patient with post-procedural pneumonia and two patients with arrhythmia and hypotension during the procedure, attributed to higher PEEP or tidal volumes or to anesthetic drugs, have also been described [28,52].

## 7. Conclusions

RB is a promising novel bronchoscopic technique that aims to overcome the limitations of conventional bronchoscopy and improve upon the current techniques of guided bronchoscopy for the investigation of pulmonary nodules. Despite the lower complication rate, current evidence suggests a lower diagnostic yield compared to TTNB. Additional prospective studies with uniform definition of efficacy metrics are required to adequately evaluate the role of RB in the diagnosis of pulmonary nodules.

## Figures and Tables

**Figure 1 cancers-16-01179-f001:**
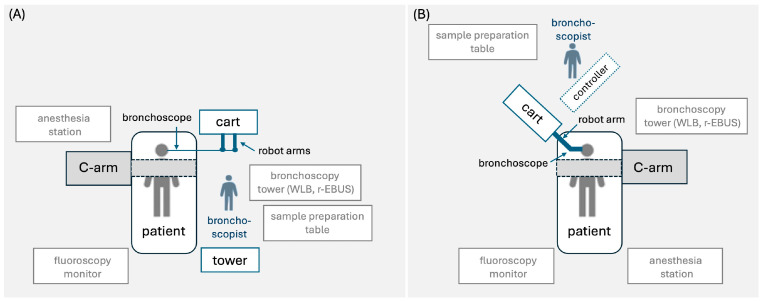
Proposed room setup for robotic bronchoscopy platform consisting of (**A**) tower, cart, bronchoscope, and handheld controller (Monarch platform), and (**B**) cart and bronchoscope with (Galaxy platform) or without (Ion Endoluminal platform) handheld controller. The schematic room setup shows the positions of the patient, bronchoscopist, hardware, robot arm(s), C-arm, bronchoscopy tower with white-light bronchoscope and radial-EBUS probe, fluoroscopy monitor, and anesthesia station [20,22].

**Figure 2 cancers-16-01179-f002:**
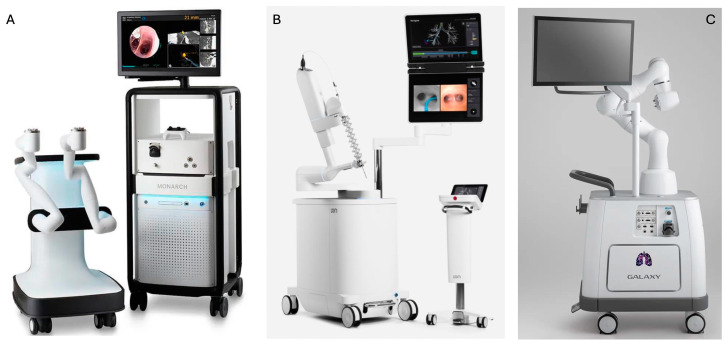
Pictures of the (**A**) Monarch platform, (**B**) Ion Endoluminal platform, and (**C**) Galaxy platform [31].

**Table 1 cancers-16-01179-t001:** Comparison of technical characteristics among robotic bronchoscopy platforms.

Characteristic	Monarch	Ion Endoluminal	Galaxy
Navigation technology	Electromagnetic navigation	Shape-sensing	Electromagnetic navigation
Bronchoscope OD	6.0 mm outer sheath, 4.2 mm inner catheter	3.5 mm	4.0 mm
Working channel diameter	2.1 mm	2.0 mm	2.1 mm
Bronchoscope flexion	outer sheath 130°, inner catheter 180°	180°	
Bronchoscope re-using	Yes	Yes	No (single-use bronchoscope)
Vision during navigation	Yes	Yes (using a 1.7 mm OD vision probe through the working channel)	Yes
Vision during sampling	Yes	No	Yes
Controller	Video game-like handheld controller	Trackball and scroll-wheel controller	Video game-like handheld controller
Integrated imaging modalities	CBCT, fluoroscopy	CBCT, fluoroscopy	Digital tomosynthesis, tool-in-lesion technology (TiLT)

OD: outer diameter, CBCT: cone beam computed tomography.

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
