# Peer review of "Robotic Bronchoscopy in Lung Cancer Diagnosis"

_cancers, 2024, doi:10.3390/cancers16061179_

Round 1

Reviewer 1 Report

Comments and Suggestions for Authors

The paper provides a comprehensive overview of robotic bronchoscopy (RB) in the context of diagnosing lung cancer, highlighting its emergence as a solution to the limitations of conventional bronchoscopy, especially in accessing distal airways to evaluate pulmonary nodules. It meticulously outlines the rationale, methods, and findings regarding the efficacy and safety of RB, alongside comparing various FDA-approved RB platforms. The review is well-structured and informative.

Additional photographs of the robotic equipment would provide a better understanding of the equipment.

Author Response

Dear Sir/Madam,

Thank you for your kind comments. In response to your comments, we state the following:

C1: Additional photographs of the robotic equipment would provide a better understanding of the equipment.

R1: Two figures were added. Figure 1 showing a schematic room setup of the robotic bronchoscopy platforms and Figure 2 showing pictures of the robotic bronchoscopy platforms.

Reviewer 2 Report

Comments and Suggestions for Authors

In general this is a well written and comprehensive review- however a few suggestions

1. a methodology of how you selected and identified the papers for inclusion

2. no figures are included- given the fact that the journal is not a specialist pulmonary journal, a figure of the components of a robotic system (patient, hardware, EMN platform, robot arm, options for cone beam, fluoroscopy, rpEBUS

3. do the authors have an opinion regarding GA/paralysis/ conscious sedation?

4. I may have missed it but should the authors mention the advantage of staging the mediastinum with a bronchoscopic as opposed to a transthoracic approach?

5. the authors may want to mention prediction models (Brock/Herder) and the potential of liquid biopsy in introduction

6. for completeness BTPNA may merit a mention

Comments on the Quality of English Language

there are a number of areas where language quality needs to be assessed.

one recurring comment is the length of many sentences- for example lines 75-91 contain 3 sentences in total- they are not clear and need to be broken down. similarly for line 52-57. Fir the structure of this article a rule of thumb might be to review any sentence that goes over 2 lines.

there are other grammatical errors I have identified, I am sure there will be others identified by journal when final review occurs- for example:

1. abstract- line 24 take our word "rather"

2. introduction- line 39- remove the from why the efforts. remove for this purpose from line 43

3. benignity is not a word I have ever read in any journal

4. line 71 should start with "The aims"

5. the descriptions of the robotic systems are very crisp- I presume you have made sure that they are not directly from the manufacturer.

6. table 2- I would take term "human patients" out of title just state human studies

Author Response

Dear Sir/Madam,

Thank you for your kind comments. In response to your comments, we state the following:

Comments and Suggestions for Authors

C1: A methodology of how you selected and identified the papers for inclusion.

R1: Two sentences describing the methodology of identifying articles used for this manuscript, were added at the end of the Introduction.

C2: no figures are included- given the fact that the journal is not a specialist pulmonary journal, a figure of the components of a robotic system (patient, hardware, EMN platform, robot arm, options for cone beam, fluoroscopy, rpEBUS

R2: Two figures were added. Figure showing a schematic room setup of the robotic bronchoscopy platforms and Figure 2 showing pictures of the robotic bronchoscopy platforms

C3: do the authors have an opinion regarding GA/paralysis/ conscious sedation?

R3: we are against the regular use of paralysis and therefore the word “rarely” was added in the sentence “Neuromuscsular blockade may rarely be used to suppress patient movements and cough…”

C4: I may have missed it but should the authors mention the advantage of staging the mediastinum with a bronchoscopic as opposed to a transthoracic approach?

R4: a sentence mentioning that bronchoscopy has the advantage of staging the mediastinum as opposed to TTNB, was added in the first paragraph of Section 2.

C5: the authors may want to mention prediction models (Brock/Herder) and the potential of liquid biopsy in introduction.

R5: The Brock/Herder model was added in the Introduction

C6: for completeness BTPNA may merit a mention.

R6: BPTNA was added in the non-surgical methods of nodule sampling in Section 2.

Comments on the Quality of English Language

C1: There are a number of areas where language quality needs to be assessed. One recurring comment is the length of many sentences- for example lines 75-91 contain 3 sentences in total- they are not clear and need to be broken down. similarly for line 52-57. Fir the structure of this article a rule of thumb might be to review any sentence that goes over 2 lines.

R1: Suggested sentences and several other long sentences were broken down.

C2: there are other grammatical errors I have identified, I am sure there will be others identified by journal when final review occurs- for example:

  1. abstract- line 24 take our word "rather" – R: the word “rather” was eliminated
  2. introduction- line 39- remove thefrom why the efforts. remove for this purposefrom line 43. - R: the word “the” and the phrase “for this purpose” were eliminated
  3. benignity is not a word I have ever read in any journal – R: the word “benignity” was replaced by “non-malignancy”
  4. line 71 should start with "The aims" – R: The word “Aim” at the beginning of the last paragraph of the Introduction, was replaced by “The aims”
  5. the descriptions of the robotic systems are very crisp- I presume you have made sure that they are not directly from the manufacturer. R: The descriptions of the robotic platforms were not taken directly from company websites, brochures or manuals. They were based on published articles, included in the reference list of the present manuscript.
  6. table 2- I would take term "human patients" out of title just state human studies – R: the term “human patients” was eliminated, and the term “human studies” was added at the beginning of the title.

Reviewer 3 Report

Comments and Suggestions for Authors

Dear Authors,

I read your article with great attention. Robotic bronchoscopy will certainly develop, so it is important to familiarize yourself with this topic.

My comments are small remarks.

1. I would like to see a paragraph in the work in which the authors attempt to describe in which group the use of robotic bronchoscopy is proposed.

2. I did not find detailed information in the work about the types of tools used (taking material). Are they forceps, a needle (what kind of needle), or a cytological brush.

3. Maybe the authors can show where, in what countries and how many such systems are used.

4. The Galaxy system was not included in table no. 2. Why? I have the impression that the system was only tested on animals? This should be explained somewhere in the text.

Author Response

Dear Sir/Madam,

Thank you for your kind comments. In response to your comments, we state the following:

C1: I would like to see a paragraph in the work in which the authors attempt to describe in which group the use of robotic bronchoscopy is proposed.

R1: A paragraph describing the patient group that would benefit from the use of robotic bronchoscopy, has been added at the end of Section 2.

C2: I did not find detailed information in the work about the types of tools used (taking material). Are they forceps, a needle (what kind of needle), or a cytological brush.

R2: A sentence about the types of tools that can be used has been added in Section 3: “Although each robotic platform has its own needle (i.e., Auris needle and Flexision needle) for nodule sampling, other needles (e.g. 21G Olympus TBNA needle, 19G or 21G Intuitive Surgical needle) and biopsy tools (e.g., Endojaw Olympus forceps, 1.1mm Erbe Tuebingen cryoprobe) can also be used”.

C3: Maybe the authors can show where, in what countries and how many such systems are used.

R3: The number of robotic systems that are currently used globally, is beyond our knowledge. However, we added a new column in Table 2 showing the country where each study was performed.

C4: The Galaxy system was not included in table no. 2. Why? I have the impression that the system was only tested on animals? This should be explained somewhere in the text.

R4: In the fourth paragraph of Section 5, it is stated that there is only one ongoing human study evaluating the Galaxy system. Preliminary results of the first human study (FRONTIER study) with the Galaxy system were added in Table 2.